# An Efficient One-Shot Federated Medical Imaging via Variational Inference Parametric Feature Transfer

## Abstract

This study introduces a one-shot federated technique for medical imaging called FBPFT-VI, a Variational Inference parametric feature-transfer approach. Each client freezes an Attention-MobileNetV2 encoder to extract features, then fits a variational posterior over its class-conditional feature statistics and transmits only the posterior parameters. The server samples synthetic features from these posteriors and trains a cosine classifier head, using Variational Inference to combine client contributions in a single aggregation round. Across multiple medical imaging benchmarks under IID and heterogeneous settings, FBPFT-VI improves the communication–accuracy trade-off.

## 1 Introduction

Federated learning (FL) enables collaborative model training without centralizing data by keeping raw samples on devices and exchanging only model updates with a server (Konecný et al., 2016; Kairouz et al., 2021). Although multi-round algorithms such as FedAvg and FedProx (McMahan et al., 2016; Li et al., 2020a) achieve strong performance, they incur heavy communication and synchronization costs, suffer from straggler and security issues, and increase the attack surface (Jhunjhunwala et al., 2024). One-shot FL mitigates these limitations by performing a single aggregation round (Guha et al., 2019; Wang et al., 2025), yet existing approaches based on knowledge distillation (Gong et al., 2021; Li et al., 2020b; Zhang et al., 2022; Jhunjhunwala et al., 2024) or neuron-matching/model-fusion (Ainsworth et al., 2022; Entezari et al., 2021; Choshen et al., 2022; Jin et al., 2022) remain sensitive to client heterogeneity and often depend on public data or exposed features. Recent progress in parametric feature transfer (PFT) (Beitollahi et al., 2024) addresses this by summarizing client feature distributions with compact probabilistic models for server-side synthetic training. We propose **FBPFT-VI**, a variational inference, one-shot FL framework in which each client fits a variational posterior over class-conditional features from a neural network. These posteriors capture epistemic uncertainty and transmit only variational parameters, preserving privacy and efficiency. The server then samples synthetic embeddings from the aggregated variational posteriors to train a cosine classifier head, achieving superior accuracy and robustness under IID and heterogeneous medical imaging scenarios[1].

## 2 Methodology

We introduce a one-shot FL model that aggregates an encoder neural network once and trains only a server-side cosine classifier on synthetic features, never sharing raw samples. Each client $u_k$ with a local dataset $\mathcal{X}_k$ optimizes an embedding network $f_{\theta_k} : \mathbb{R}^{H \times W \times C} \to \mathbb{R}^d$ so that normalized latent

---

[1]The acknowledgment section has been omitted for the double-blind review process and will be included in the final version of the paper.

Submitted to 39th Conference on Neural Information Processing Systems (NeurIPS 2025). Do not distribute.

Table 1: Comparative IID Performance (%) on Eight Datasets. The best results for each dataset are highlighted in **bold**, and the second-best results are underlined.

| Method | Blood | Derma | Oct | Path | Tissue | RSNA | Diabetic | ISIC |
|---|---|---|---|---|---|---|---|---|
| FedAvg (McMahan et al., 2016) | 93.51 | 74.61 | 75.60 | 84.54 | 63.64 | 88.16 | 49.04 | 62.88 |
| FedAvg(1) | 13.74 | 66.88 | 25.00 | 5.86 | 32.07 | 78.65 | 35.60 | 38.05 |
| DAFL (Chen et al., 2019) | 7.13 | 66.43 | 25.00 | 7.63 | 11.55 | 50.55 | 22.63 | 14.51 |
| DENSE (Zhang et al., 2022) | 39.37 | 66.93 | 33.80 | 21.89 | 21.35 | 55.06 | 23.51 | 13.69 |
| FedISCA(Kang et al., 2025) | 87.99 | 70.12 | 70.20 | 84.18 | **61.90** | 85.34 | 40.08 | 48.39 |
| E-FedISCA(Kang et al., 2025) | 87.31 | **71.47** | 71.30 | 79.48 | 57.96 | **85.46** | 41.32 | 51.17 |
| FBPFT-VI | **88.86** | 67.38 | **81.20** | **88.34** | 57.64 | 84.49 | **49.54** | **68.99** |

features $\hat{z} = z/\|z\|_2$ align with class proxies (Movshovitz-Attias et al., 2017). Specifically, with classes proxies $\mathcal{P}_k = \{\hat{p}_{k,c}\}_{c=1}^C$ for client $k$ and temperature $\tau > 0$, the local objective for a sample $x$ with class $y$ is

$$\mathcal{L}_{\text{proxy}}(\theta_k) = \mathbb{E}_{(x,y)\sim\mathcal{X}_k}\left[-\log \frac{\exp\{\langle\hat{z}, \hat{p}_{k,c}\rangle/\tau\}}{\sum_{c=1}^C \exp\{\langle\hat{z}, \hat{p}_{k,y}\rangle/\tau\}}\right], \qquad \hat{z} = \frac{f_{\theta_k}(x)}{\|f_{\theta_k}(x)\|_2}.$$

30 Our **major** novelty is to propose a variational inference to estimate the class proxies. To summarize
31 local distributions without exposing per-example embeddings, each client fits per-class mean-field
32 Gaussians in the feature space of $f_\theta$ for class $c$, $q_{k,c}(z) = \mathcal{N}(\mu_{k,c}, \Sigma_{k,c})$, $\Sigma_{k,c} = \text{Diag}(\sigma_{k,c}^2)$.
33 To enable efficient stochastic optimization over the latent proxy distributions, we apply the repa-
34 rameterization trick (Barros et al., 2024). Specifically, each proxy sample $p_{k,c}$ is obtained as
35 $p_{k,c} = \mu_{k,c} + \sigma_{k,c} \odot \epsilon$, where $\epsilon \sim \mathcal{N}(0, \mathbf{I})$. Let $\mathcal{P}$ be the client proxies and write the Boltzmann
36 likelihood $p_\theta(\mathcal{X}_k \mid \mathcal{P}) \propto \exp\{-\mathcal{L}_{proxy}(\mathcal{X}_k)/\alpha\}$ with constant $\alpha > 0$, the direct posterior inference
37 is intractable (Kingma and Welling, 2022), so we adopt a server-as-prior strategy and minimize, for
38 client $u_k$, $\mathcal{L}_k(\{\mathcal{P}_k, \theta_k\}; s) = \mathbb{E}_{q_{k,c}}\left[\mathcal{L}_{proxy}(\mathcal{X}_k)\right] + \alpha\,\text{KL}\left(q_{\phi_k}(\mathcal{P})\|\mathcal{N}(0,\mathbf{I})\right)$.

39 After local optimization, each client $k$ transmits its posterior parameters $\{\mu_{k,c}, \sigma_{k,c}\}_{c=1}^C$ to the
40 server. Instead of directly averaging weights as in FedAvg, the server combines all received proxy
41 distributions into a single *multimodal global distribution* $q_{\text{global}}(z \mid c) = \bigcup_{k=1}^K q_{k,c}(z)$, which
42 represents the ensemble of client knowledge across heterogeneous models and data. From this global
43 mixture, the server samples synthetic feature embeddings $\tilde{z} \sim q_{\text{global}}(z \mid c)$ and uses them to train a
44 one-layer cosine classifier head.

## 3 Experimentation

46 **FL settings:** We simulated an FL environment with five clients, each using a **MobileNetV2** backbone
47 with trainable convolutional layers and a 128-dimensional embedding space. Datasets were partitioned
48 following (Kang et al., 2025) into three settings: (i) Dirichlet non-IID with $\alpha = 0.3$, (ii)$\alpha = 0.6$,
49 and (iii) IID with 5 clients for balanced data. Each client trained locally for 8 epochs using Adam
50 (lr=$3\times10^{-4}$), followed by 30 epochs of head fine-tuning (lr=$10^{-3}$, weight decay=$10^{-4}$). Variational
51 inference with a diagonal Gaussian posterior was applied using 1500 steps (lr=$5\times10^{-3}$). Aggregation
52 was performed in a *single communication round*, and the resulting probabilistic embeddings were
53 used for evaluation.

54 **Results:** Table 1 summarizes the IID, and our proposed FBPFT-VI outperformed existing FL
55 baselines, achieving the best accuracy on six (out of eight) datasets. Thus, OCT and Path reached
56 81.20% and 88.34%, respectively, surpassing FedISCA (Kang et al., 2025) and E-FedISCA. Besides,
57 FedAvg if the standard multi-round version, while FedAvg(1) denotes its one-shot variant. Among the
58 one-shot methods, FBPFT-VI achieved the highest overall performance, outperforming DAFL (Chen
59 et al., 2019) and DENSE (Zhang et al., 2022). To evaluate robustness under non-IID data, we used
60 Dirichlet distributions with $\alpha = 0.6$ and $\alpha = 0.3$, as shown in Table 2. As expected, accuracy
61 decreased with stronger heterogeneity (smaller $\alpha$). Under moderate heterogeneity ($\alpha = 0.6$),
62 FBPFT-VI achieved the best performance on OCT (79.40%) and Path (89.21%), while under high
63 heterogeneity ($\alpha = 0.3$) it surpassed all baselines across four (out of five) datasets. For model
64 heterogeneity experiments (Table 3), we followed the setup of Kang et al. (2025), where clients used

Table 2: Classification accuracy (%) on five datasets with different heterogeneity levels. The best results for each dataset are highlighted in **bold**, and the second-best results are underlined.

| | Dirichlet ($\alpha = 0.6$) | | | | | Dirichlet ($\alpha = 0.3$) | | | | |
|---|---|---|---|---|---|---|---|---|---|---|
| Method | Blood | Derma | Oct | Path | Tissue | Blood | Derma | Oct | Path | Tissue |
| FedAvg (McMahan et al., 2016) | 93.60 | 72.72 | 76.50 | 81.48 | 55.61 | 87.49 | 69.88 | 73.50 | 77.52 | 53.26 |
| FedAvg(1) | 18.24 | 66.88 | 25.00 | 5.86 | 32.07 | 16.92 | 10.97 | 25.00 | 5.86 | 32.07 |
| DAFL (Chen et al., 2019) | 7.13 | 66.88 | 34.40 | 14.97 | 39.15 | 7.13 | 13.62 | 25.00 | 18.64 | 45.00 |
| DENSE (Zhang et al., 2022) | 34.52 | 67.78 | 39.40 | 30.31 | 9.47 | 30.78 | 12.77 | 25.80 | 19.87 | 9.33 |
| FedISCA (Kang et al., 2025) | 82.90 | 69.83 | 68.60 | 82.92 | 53.04 | 46.59 | 15.91 | 60.50 | 79.25 | 51.00 |
| E-FedISCA (Kang et al., 2025) | **83.10** | **69.68** | 66.20 | 78.51 | 52.87 | 45.86 | 16.96 | 61.60 | 73.40 | 48.45 |
| FBPFT-VI (Our model) | 82.17 | 63.39 | **79.40** | **89.21** | 55.51 | **83.51** | **62.24** | **76.00** | **87.08** | **56.46** |

Table 3: Classification performance (%) across five datasets under the model heterogeneity. The best results for each dataset are highlighted in **bold**, and the second-best results are underlined.

| | IID | | | | | Dirichlet ($\alpha = 0.6$) | | | | | Dirichlet ($\alpha = 0.3$) | | | | |
|---|---|---|---|---|---|---|---|---|---|---|---|---|---|---|---|
| Method | Blood | Derma | Oct | Path | Tissue | Blood | Derma | Oct | Path | Tissue | Blood | Derma | Oct | Path | Tissue |
| DAFL (Chen et al., 2019) | 7.13 | 65.69 | 25.00 | 15.72 | 35.66 | 7.13 | 67.21 | 37.10 | 28.15 | 39.54 | 7.13 | 13.47 | 45.30 | 29.68 | 19.54 |
| DENSE (Zhang et al., 2022) | 46.86 | 66.88 | 44.00 | 33.08 | 38.28 | 23.47 | 67.93 | 40.70 | 28.68 | 36.70 | 34.67 | 13.42 | 44.00 | 39.37 | 38.37 |
| FedISCA (Kang et al., 2025) | 87.96 | 71.17 | 70.00 | 83.02 | 61.74 | 73.43 | **69.23** | 64.80 | 82.73 | 51.95 | 44.20 | 16.61 | 62.00 | 72.26 | 43.80 |
| E-FedISCA (Kang et al., 2025) | 88.31 | **71.72** | 71.00 | 80.04 | 58.96 | 72.76 | **69.23** | 64.60 | 80.84 | 52.04 | 47.18 | 16.01 | 58.90 | 72.17 | 43.19 |
| FBPFT-VI (Our model) | **94.12** | 66.58 | **81.10** | **93.65** | 60.81 | **89.54** | 57.66 | **79.50** | **92.84** | **55.23** | **89.74** | **65.74** | **78.00** | **91.34** | **56.30** |

ResNet34, WRN-16-2, VGG16 (BN), and VGG8 (BN). We replaced ResNet18 with our MobileNet-Attention model for one client to represent architectural diversity better. Under this configuration, FBPFT-VI achieved superior accuracy across four (out of five) datasets, especially on Path (93.65%) and OCT (81.10%) in IID conditions, and maintained top performance even with strong heterogeneity ($\alpha = 0.3$). These results indicate that our feature-based parameter fusion effectively integrates information from clients with different model capacities. Scalability analysis (Table 4) was conducted by increasing the number of clients from 5 to 20 under IID settings. Although accuracy decreased due to increased communication diversity, FBPFT-VI remained competitive. The two-shot variant further improved performance, for instance, achieving 82.50% on OCT (while keeping communication costs low).

## 4 Conclusion

This work presented **FBPFT-VI**, a variational one-shot FL framework for medical imaging that transfers variational inference class-conditional feature distributions instead of raw data. By modeling client knowledge as variational posteriors and training a cosine classifier on sampled synthetic embeddings, FBPFT-VI achieves efficient, privacy-preserving aggregation while capturing epistemic uncertainty. Experiments on eight medical datasets show consistent improvements over both one-shot and two-round baselines under IID, non-IID, and heterogeneous settings, showing that Bayesian feature modeling offers an effective trade-off between privacy, communication efficiency, and accuracy in federated medical learning.

Table 4: Classification accuracy (%) of 20 clients across five datasets under IID settings. The best results for each dataset are highlighted in **bold**, and the second-best results are underlined.

| Method | Blood | Derma | Oct | Path | Tissue |
|---|---|---|---|---|---|
| FedISCA (20 clients) | 78.31 | 69.23 | 67.20 | 84.57 | 56.93 |
| E-FedISCA (20 clients) | 79.07 | 69.18 | 64.50 | 81.45 | 54.62 |
| E-FedISCA (20 clients) 2-shot | 86.23 | 69.48 | 69.80 | 83.19 | 58.57 |
| FBPFT-VI (20 clients) | 76.82 | 64.04 | 78.80 | 86.98 | 54.13 |
| FBPFT-VI (20 clients) 2-shot | 85.91 | 63.24 | 82.50 | 84.85 | 54.02 |

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
