# OpenReview forum: "An Efficient One-Shot Federated Medical Imaging via Variational Inference Parametric Feature Transfer"
_EurIPS.cc/2025/Workshop/MedEurIPS — EurIPS 2025 Workshop MedEurIPS Submission_

### Official Review · Reviewer_9xUX · 2025-10-24
**Review for An Efficient One-Shot Federated Medical Imaging via Variational Inference Parametric Feature Transfer**

**Rating:** 7
**Confidence:** 3

**Review:**

Summary:
This paper introduces FBPFT-VI, a one-shot federated learning method for medical imaging that uses variational inference to transfer class-level feature distributions instead of raw data. It achieves better accuracy and robustness than existing approaches while maintaining privacy and communication efficiency.

Strengths:
- Interesting and well-motivated idea combining federated learning with medical imaging.
- Well-designed experimental setup covering IID, non-IID, and heterogeneous scenarios.

Suggestions:
- Clarify experimental details (e.g., dataset splits, repetitions) to ensure reproducibility.
- Provide quantitative analysis of privacy and communication efficiency.

---

### Official Review · Reviewer_x1tH · 2025-10-30
**One-Shot Federated Learning for Medical Imaging using Variational Inference Feature Transfer**

**Rating:** 6
**Confidence:** 4

**Review:**

`Overview:`
The paper proposes FBPFT-VI, a novel one-shot federated learning (FL) method tailored for medical imaging. The core idea is to avoid costly multi-round communication by having each client fit and transmit only the parameters of a variational posterior (class-conditional mean-field Gaussians) that describe their local feature distributions. The server then aggregates these distributions, samples synthetic features, and trains a simple classifier head in a single round.

- This work is relevant in tackling the critical challenges of communication efficiency and privacy in federated medical imaging. The novelty lies in applying variational inference (VI) to the parametric feature transfer paradigm. Using VI to model feature distributions and capture epistemic uncertainty is an interesting approach to aggregating knowledge from heterogeneous clients, which is a key bottleneck in one-shot FL.
- The experimental results are strong and well-structured for a short paper. The method shows strong performance relative to several baselines (including multi-round FedAvg and other one-shot methods) across eight medical datasets. Its robustness in non-IID and model-heterogeneous settings is particularly promising.

`Suggestions:`
Given the space constraints, the methodology is very compressed. The relationship between the initial local encoder optimization (using $\mathcal{L}_{proxy}$ and the subsequent fitting of the variational posterior could be slightly clearer. A brief clarification on this workflow (e.g., "clients first train encoders, then freeze them to extract features and fit posteriors") would aid readability. The paper claims "preserving privacy" while transmitting feature statistics is a significant improvement over raw data or features; it would be beneficial to briefly acknowledge any potential, non-trivial privacy leakage that might still exist from these statistics.

`Conclusion:`
The proposed method is novel and the supporting results are strong, making it an excellent fit for the workshop. It has the potential to stimulate valuable discussion on efficient and privacy-aware FL.

`Score:` 4 Weak accept.

---

### Decision · Program_Chairs · 2025-10-31

**Decision:**

Accept (Poster)

**Comment:**

Both reviewers find the paper well motivated and relevant.